# Modeling RTT Syndrome by iPSC-Derived Neurons from Male and Female Patients with Heterogeneously Severe Hot-Spot MECP2 Variants

**DOI:** 10.3390/ijms232214491

**Published:** 2022-11-21

**Authors:** Sara Perego, Valentina Alari, Gianluca Pietra, Andrea Lamperti, Alessandro Vimercati, Nicole Camporeale, Maria Garzo, Francesca Cogliati, Donatella Milani, Aglaia Vignoli, Angela Peron, Lidia Larizza, Tommaso Pizzorusso, Silvia Russo

**Affiliations:** 1Experimental Research Laboratory of Medical Cytogenetics and Molecular Genetics, IRCCS Istituto Auxologico Italiano, Via Ariosto 13, 20145 Milan, Italy; 2Laboratory of Biology BIO@SNS, Scuola Normale Superiore, Piazza dei Cavalieri, 7, 56126 Pisa, Italy; 3Institute of Neuroscience, National Research Council (CNR), Via Giuseppe Moruzzi, 1, 56124 Pisa, Italy; 4Fondazione IRCCS Ca Granda Ospedale Maggiore, Policlinico, Via Francesco Sforza, 28, 20122 Milan, Italy; 5Department of Health Sciences, Università degli Studi di Milano, Struttura Complessa di Neuropsichiatria dell’Infazia e Adolescenza ASST GOM Niguarda, Piazza Ospedale Maggiore, 3, 20162 Milan, Italy; 6Child Neuropsychiatry Unit—Epilepsy Center, Department of Health Sciences, ASST Santi Paolo e Carlo, San Paolo Hospital, Università Degli Studi di Milano, 20142 Milan, Italy

**Keywords:** Rett syndrome, hot-spot *MECP2* pathogenic variants, X inactivation mosaicism, genotype-phenotype correlation, iPSC-neurons, early morphological neuronal biomarkers, mature neuron e-recordings

## Abstract

Rett syndrome caused by *MECP2* variants is characterized by a heterogenous clinical spectrum accounted for in 60% of cases by hot-spot variants. Focusing on the most frequent variants, we generated in vitro iPSC-neurons from the blood of RTT girls with p.Arg133Cys and p.Arg255*, associated to mild and severe phenotype, respectively, and of an RTT male harboring the close to p.Arg255*, p.Gly252Argfs*7 variant. Truncated MeCP2 proteins were revealed by Western blot and immunofluorescence analysis. We compared the mutant versus control neurons at 42 days for morphological parameters and at 120 days for electrophysiology recordings, including girls’ isogenic clones. A precocious reduced morphological complexity was evident in neurons with truncating variants, while in p.Arg133Cys neurons any significant differences were observed in comparison with the isogenic wild-type clones. Reduced nuclear size and branch number show up as the most robust biomarkers. Patch clamp recordings on mature neurons allowed the assessment of cell biophysical properties, V-gated currents, and spiking pattern in the mutant and control cells. Immature spiking, altered cell capacitance, and membrane resistance of RTT neurons, were particularly pronounced in the Arg255* and Gly252Argfs*7 mutants. The overall results indicate that the specific markers of in vitro cellular phenotype mirror the clinical severity and may be amenable to drug testing for translational purposes.

## 1. Introduction

*MECP2* gene (MIM #300005), mapping at Xq28, encodes the transcription regulator and chromatin remodeler methyl-CpG-binding protein 2 (MeCP2), which is ubiquitously expressed, but at particularly high levels in post-mitotic neurons in the brain [1,2,3,4]. Due to the key role of *MECP2* in the function of neuronal cells, pathogenic variants in heterozygous females and rarely in hemizygous males are the most common cause of the rare (1:10,000 live births) neurodevelopmental Rett syndrome (RTT MIM #312750). RTT is characterized by early developmental regression leading to loss of language, purposeful use of the hands, onset of typical stereotypies, gait abnormalities, seizures, apnea, hyperventilation, and autonomic dysfunction [5]. A further common feature in RTT patients is post-natal microcephaly, also observed in animal models, where the reduced brain volume has been associated with abnormal morphology of neurons, which show reduced dendritic branching, soma size, spine density, and synapse number [6,7].

To date, 3924 females and 345 males with *MECP2* pathogenic variants have been reported [8] outlining high male embryonic lethality or early post-natal death.

Eight hot-spot mutations (R106W, R133C, T158M, R168*, R255*, R270*, R294*, and R306C) account for more than 60% of typical RTT cases [9]. A consistent clinical heterogeneity with a strong genotype-phenotype correlation is reported in the literature indicating that early-truncating pathogenic variants, such as R168*, R255*, R270*, and large deletions are associated with a more severe phenotype than R294*, R133C, R306C, and late-truncating C-terminal variants [10]. *MECP2* being subject to X chromosome inactivation (XCI), further clinical expressivity is determined in RTT females by the variable ratio of the two cell populations expressing the wild-type (WT) and the mutant (Mut) allele [11].

Similarly to all neurodevelopmental disorders the study of RTT has been challenging for a long time due to the inaccessibility of the central nervous system and quantity and quality limitations of post mortem human brain samples [12]. Engineered mouse models have been useful to study RTT [13,14,15] despite poor translation of molecular observations into the clinic. Furthermore, the inter-species differences in brain development, function, and cell composition at fetal and adult stages [16,17] question the fidelity of the animal studies in modeling human neurodevelopmental disorders.

An avenue to understand RTT pathogenesis and disclose potential actionable therapeutic targets has been opened by patient-specific models, generated by induced human pluripotent stem cells (iPSCs) [18] differentiated to 2D neural cell cultures [19,20,21] or brain organoids [17,22,23].

To date, several iPSC lines harboring a range of *MECP2* mutations have been generated [21], mainly reprogrammed from patient-derived fibroblast cells. Most studies have employed iPSC lines from female patients, with the exception of a few male patient-specific iPSCs presenting the Q83* or the N126I mutation [24,25]. Several research groups have demonstrated that neuronal models from RTT girls showed morphological defects, such as fewer synapses, smaller soma size, reduced dendritic branching, and functional anomalies ranging from reduced cell capacitance and altered calcium signaling to defective firing activity and excitatory/inhibitory imbalance [11,19,26,27,28]. Conversely, morphological parameters appear so far inconsistent across iPSC-derived neurons (i-neurons) from male RTT patients. No differences were observed in two male cell lines harboring Q83* and N126I variants compared to controls [29], while the average neurite length was reported to be higher in control cells than in Q83* neurons [30].

Aiming at investigating whether the clinical heterogeneity observed in the patients with different hot-spot pathogenic variants may be mirrored in their cellular phenotype and to identify biomarkers tailored on specific variants, we compared iPSCs from a male with Gly252Argfs*7, two girls with the nearby Arg255* hot-spot variant, and a female with the Arg133Cys variant, this latter associated with mild clinical manifestations. According to the *MECP2* Variation Database (RettBASE), RettSyndrome.org (http://mecp2.chw.edu.au/ (accessed 1 June 2022) these three variants are representative of 18.7% of RTT cases.

We generated from peripheral blood mononuclear cells (PBMC) of *MECP2* defective girls isogenic WT or Mut iPSCs clones, then differentiated into prefrontal cortical neurons. iPSC clones from the RTT male and isogenic WT and Mut clones from the females were compared to sex-matched healthy controls to appoint morphological biomarkers in the derived young neurons and electrophysiological biomarkers in the mature neurons. Our findings reveal graded neuronal maturation deficits, informing on appropriate testing and individualized efficacy of candidate drugs.

## 2. Results

### 2.1. MECP2 Patients Selected for In Vitro Modeling

Out of 169 RTT girls with *MECP2* pathogenic variants, we have diagnosed so far, 11 carriers of p.Arg168*, 15 carriers of p.Arg255*, and eight of p.Arg133Cys representing 20% of our cohort. Aiming at modelling hot-spot pathogenic variants to check whether the in vitro neurons might reflect the clinical heterogeneity, we collected blood from two girls with p.Arg168*, six with p.Arg255* and three with p.Arg133Cys. A boy with severe encephalopathy, carrier of the p.Gly252Argfs*7 variant close to p.Arg255*, was also included in the study. A schematic of the location of the selected *MECP2* variants at the gene and protein level is shown in Figure 1.

DNAs from female lymphocytes were investigated for XCI inactivation by AR locus (Figure 2a) and DXS6673E marker in the non informative cases. A selective skewing of the *MECP2* mutated allele was observed in all the girls with p.Arg168* and in 4/6 with p.Arg255*, while all 3/3 patients with p.Arg133Cys were not selectively skewed. The cases with balanced XCI were reprogrammed to iPSCs obtaining a minimum of five clones; also lymphocytes from two patients with unbalanced XCI (PtA3 and PtC1) were reprogrammed, but no mutant clones were obtained (Figure 2a).

cDNA sequencing to disclose whether each clone expressed either WT/MUT or both alleles, confirmed the occurrence of isogenic clones expressing the variant or wild-type allele in the girls with balanced XCI (Figure 2b), while only the wild-type allele was expressed in the skewed cases. As regards the RTT male carrier of p.Gly252Argfs*7, two clones (PtY1 and PtY2), out of several obtained by reprogramming peripheral blood cells to iPSCs, were selected for this study. In parallel to RTT control, iPSC clones were obtained and characterized from four healthy males and four healthy females (CTRLs).

### 2.2. Patients Clinical Phenotype

All the modeled RTT females were referred for clinical diagnosis and follow-up at the Regional Epilepsy Center of Ospedale Santi Paolo e Carlo (Milan, Italy), while the boy was referred to our lab by the Pediatric Unit of Clinica De Marchi (Milan, Italy) and his clinical history was described in detail [31].

Table 1 reports the main clinical features of our patients. The RTT clinical severity scale (CSS) [9], indicated in the last column, confirms the major severity of the patients harboring the MeCP2 protein truncated at the NLS region. PtA2 appears very impaired, with absence of eye contact and a never acquired deambulation. PtA1, with the same variant, is still a child, and preserves eye contact and deambulation; epilepsy occurred not so early but is drug-resistant. Patients B1, B2, B3 are currently adults, manifest a controlled epilepsy, preserve independent walking and eye contact and, one of them, PtB1, is able to say single words. Microcephaly is referred for patients A1, A2, and B1.

The clinical data of the male are reported in the last line of Table 1: decreased fetal movements and fetal growth restriction were described during pregnancy; he suffered from severe hypotonia and psychomotor retardation and died at 1 year and 3 months from the severe and irreversible worsening of the encephalopathy.

### 2.3. From iPSCs to Neuron Differentiation

RTT- and CTRL-iPSC stemness was confirmed by immunostaining of the pluripotency markers OCT3/4 and Tra-1-60 (Appendix A) and expression of OCT3/4, SOX2, and NANOG by RT-PCR (Appendix A). Genomic stability of iPSCs as compared to donor blood was assessed for each clone by karyotyping (Appendix A) and SNP-array (data not shown) to exclude cytogenetic and submicroscopic rearrangements which might have occurred during in vitro culture. The images are representative of all clones analyzed. Expression of the proper *MECP2* variant was proved by DNA sequencing of all clones (Figure 2b for females and Appendix A for male).

Based on the XCI data, iPSC isogenic clones from the two patients with p.Arg255* (named PtA1 and PtA2) and from the girl with p.Arg133Cys (PtB1) were obtained. We differentiated and analyzed two independent clones (PtY1 and PtY2) of the male patient, two WT and two Mut clones of PtA1 female patient, and one WT and one Mut clone of PtA2. For the patients PtB2 and PtB3 only mutated clones were differentiated.

All the cortical i-neurons (iPSC-derived neurons) were investigated for morphological and functional biomarkers.

### 2.4. MeCP2 Protein in RTT Patients

To compare the expression of MeCP2 in the WT and mutated mature neurons, nuclear extracts at 100 days were analyzed by Western blot (WB) using N-ter (Figure 3a) and C-ter (Figure 3c) antibodies.

A lower molecular weight (MW) band of about 37kDa was detected using N-ter antibody in neurons from PtY and from mutant clones expressing p.Arg255* (PtA1 Mut), proving that the truncated MeCP2 isoforms are detectable (Figure 3a); hybridization with the C-ter antibody did not display signals for PtY and PtA1 Mut (Figure 3c), PtA2Mut (data not shown) confirming truncation of the protein in these cells. Neurons from controls, WT isogenic and missense p.Arg133Cys clones (PtB1) showed the band expected for the full length protein (Figure 3c). To investigate whether the truncated isoforms are correctly located in the nucleus, immunofluorescence (IF) with the same Abs was performed proving their proper nuclear localization and confirming the lack of signal in the cases with truncated isoforms (Figure 3b,d).

### 2.5. Morphological Analyses of RTT Young Neurons

With the aim of comparing the in vitro neuronal phenotype of RTT patients to their different degree of clinical severity, we investigated some morphological hallmarks of RTT pathology in the i-neurons from the patients selected for hot-spot pathogenic variants. At the 42th day, a week after the stage of progenitors (35 days of neuronal differentiation), a restricted nuclear area size was observed in the i-neurons from the two independent PtY1 and PtY2 clones of the male patient expressing p.Gly252Argfs*7, and from the mutated clones of all RTT girls. Figure 4a provides a representative overview of neurons of PtY, PtA (p.Arg255*), and PtB (p.Arg133Cys) matched to 42 days healthy control neurons.

A comparison between the nuclear area of the pooled data of the two clones of PtY, and all the mutant clones of the RTT females evidenced a significant nuclear size reduction with the exception of PtB1 and PtB3, carriers of a variant associated to a milder phenotype (Figure 4b). As shown in Figure 4c the areas of each isogenic mutant vs. its WT clone are significantly different in carriers of p.Arg255*, whereas they do not differ in those with p.Arg133Cys. The analysis of nuclear area size among the isogenic controls did not evidence significant differences (Figure 4e), while the comparison between WT isogenic clones and the pooled healthy controls data revealed a significant difference (Figure 4d).

At the same precocious neuronal differentiation stage, we analyzed dendritic complexity and neuronal length, which have been reported as defective in mature neurons [32,33,34]. An overview of the morphological appearance of CTRL, PtY, PtA1, PtA2, and PtB1 neurons is provided in Figure 5a, while panels b, c, and d summarize the results of the statistical analyses. The i-neurons from all the clones expressing p.Arg255* (PtA1 Mut and PtA2 Mut) appear to have the same morphological abnormalities of PtY whose MeCP2 protein is altered from the 252 aa residue.

The box-plots in Figure 5 display a significant decrease in the number of end points b and branches c, and a reduced neuronal length d in all mutant clones. Appendix A shows the data from the two clones of PtY (p.Gly252Argfs*7) and PtA1 (p.Arg255*), pointing out that the technical replicates behave similarly in almost all parameters, with significant deregulation when compared to controls. Less severe alterations were evidenced for i-neurons with p.Arg133Cys missense mutation (PtB1 Mut, PtB2 Mut, and PtB3 Mut). Four comparisons were made for each parameter (Figure 5b–d). All mutant clones were compared with a pool of healthy controls (upper-left box-plots). PtA1, PtA2, and PtB1 were also compared with their specific isogenic WT clones (bottom-left). No statistical significance (ns) was found among the WT clones (bottom-right graphs) which also appeared similar to the CTRL pool (upper-right).

### 2.6. Electrophysiology of MECP2 Defective Cortical Neurons

To assess the maturity of mutant and wild-type cells at functional level, we used whole cell patch-clamp to record firing properties and voltage-gated currents. Spiking was induced by applying a series of incremental current steps in current clamp mode. This protocol tests the mature neurons’ (120 days) capability to fire multiple action potentials (APs) as a sign of their maturation state. Cell firing pattern was divided into four categories: A1 cells unable to trigger APs; A2 cells firing a single action potential; A3 cells generating a series of APs but not able to regenerate completely this capability between an AP and the following one, determining a decreasing amplitude of each AP and, in some cases, a silencing of the neuron until the end of the current step; A4/A5 representing a mature neuron firing tonically and phasically, respectively. Typical traces for each category are depicted in Figure 6a. By comparing the frequency distributions of a pool of controls cells (Pool CTRLs) with p.Gly252Argfs*7 mutant cells and two different case carriers of the p.Arg255* variant, we observed (Figure 6b) a significant shift in the distribution towards a less mature phenotype rich in A1, A2, and A3 classes and with fewer A4–A5 cells (X square test: PtY *p* < 0.01, PtA1 Mut *p* = 0.02, PtA2 Mut *p* < 0.01). The comparison was performed with respect to a pooled population of healthy control clones, not including isogenic clones, that were shown to be not significantly different (Appendix A).

The Gly252Argfs*7 mutation also induced both reduced cell capacitance and increased membrane resistance (Mann–Whitney *p* < 0.05, Figure 6c). The p.Arg255* variant showed a reduction in cell capacitance and action potential threshold (Figure 6c). Comparison of patient PtA2 with its isogenic control confirmed the reduced cell capacitance, but it also showed increased membrane resistance and resting potential. However, these differences were present only in one of the patients suggesting a possible involvement of the genetic background (Figure 6c,e). The p.Arg133Cys cells displayed a different phenotype with increased cell capacitance and a minimal, although statistically significant, reduction in membrane resistance (Figure 6i). Overall, these results underscore the strong impairment present in the male Gly252Argfs*7 mutant and the specific phenotype of the p.Arg133Cys variant.

To strengthen our data, we also performed the following comparisons. First, considering that the cells carrying the Gly252Argfs*7 mutation derived from a male patient, we compared it with a control male group (Figure 6f,g). The two distributions were significantly different (Mann–Whitney test, PtY vs. CTRL-Y *p* < 0.01) confirming the strong phenotype present in cells with this mutation. Then, in Figure 6d,e we compared one of the two cases carrying the Arg255* mutation with its isogenic WT clone confirming the different firing pattern induced by the mutation (Mann–Whitney test, PtA2 Mut vs. Isogenic WT *p* < 0.01). Third, we exploited the availability of the isogenic control for the PtB1 patient to test whether the spiking behaviour and other functional properties of PtB1 cells are also different from isogenic controls. The chi-square test revealed a significantly different distribution of spiking pattern (*p* < 0.01, Figure 6h), and cell capacitance and membrane resistance (*p* < 0.05 Figure 6i).

Finally, in Supplementary Figure 3c,d we compared data of two different clones derived from the same patient with the mutation Gly252Argfs*7. No statistical difference was present in the firing frequency and in the other functional properties between these groups (PtY1 vs. PtY2 *p* > 0.05) confirming the robustness of this phenotype.

In some of the clones we also characterized possible differences in the voltage gated currents underlying the action potential. The results, shown in Supplementary Figure 4, revealed differences for the delayed V-dip K^+^ current of the Gly252Argfs*7 mutation (as compared to the CTRL pooled cells), while the Arg255* (PtA1 Mut compared to the pooled CTRL cells) mutation and the Arg133Cys mutation (compared to the isogenic control) were characterized by an altered V-dip Na^+^ current.

## 3. Discussion

The human brain is notably more complex than the brain of mouse and rat of which engineered models have yielded crucial information on the comprehension of neurodevelopmental human disorders, even if with the limitation of difficult or incomplete translation to the clinic [35]. The iPSC-based technology, suitable to generate patient-specific neuronal models has provided a challenging opportunity to unravel the kind of disorders focusing on the human brain cells along the differentiation pathway while also capturing intra-patient clinical expressivity. During the last ten years, human stem cell-based neuronal models have been generated from fibroblasts of patients with several *MECP2* pathogenic variants, enabling the knowledge of the causative mechanisms, the pathophysiology of RTT syndrome, and the identification of actionable therapeutic drugs [21]. However, only a small number of iPSC clones from girls carriers of a few hot-spot *MECP2* variants has been characterized, and to our knowledge, none of the clones were obtained from blood [20]. Our study focused on p.Arg255* and p.Arg133Cys, two hot-spot *MECP2* variants recognized as representative of moderate and mild phenotypical expression of the disease, respectively. Because taking a blood sample is a simpler and less invasive procedure than skin biopsy, we generated iPSC clones from blood. Unexpectedly a skewed XCI leading to the selective expression of the wild-type alleles was monitored in 6/8 (75%) girls, carriers of variants associated with the more severe clinical phenotype, as p.Arg168* and p.Arg255* [9]. This finding points to the deviation from the balanced XCI mosaicism which should be carefully explored to comprehend its role in modulating the clinical expression of Rett syndrome. In the context of in vitro modeling the XCI potential bias also deserves to be kept under control for all X-linked disorders in order to interpret correctly the results on the prevalent female cases.

By selecting for terminal differentiation only the randomly X inactivated cases, our study succeeded in generating RTT in vitro cortical neurons from both the mutant and the isogenic iPSC clones of two girls with p.Arg255* variants and two with p.Arg133Cys. A male carrier of p.Gly252Argfs*7 variant truncating the protein close to the p.Arg255* female mutation was modelled for cross comparison of the respective MeCP2-defective neurons. To ascertain the effective deficit of MeCP2 in our patients’ i-neurons, we proved by the combined use of N-ter and C-ter antibodies, the presence of a truncated MeCP2 protein only in the neurons derived from the clones expressing the variant and excluded a phenomenon of non-sense mediated RNA decay. To our knowledge this is the first protein assay on MeCP2-defective neurons, demonstrating truncated proteins. Further, using the same antibodies in immunofluorescence, we observed that the MeCP2 protein encoded by the missense variant is properly localized within the nucleus and the truncated MeCP2 proteins, despite terminating at the NLS, can reach the nucleus. In addition, the C-ter Ab allowed follow-up of the expression of the WT MeCP2 whose presence may be considered a potential biomarker in validating the efficacy of drugs administered to reactivate the WT allele.

As the association between the heterogenous clinical expressivity of a neurological disease and the cellular biomarkers observed in the patient’s i-neurons might be a helpful tool to monitor the real efficacy of a treatment, the clinical histories of our patients were revisited and aligned to the severity of their in vitro neuronal phenotype. In keeping with the literature [9] the three girls with the missense p.Arg133Cys all displayed a lower clinical severity scale, with regression occurring after 2.5 years, maintenance of the autonomous walking in adult age, and drug-controlled seizures. Conversely PtA2, a girl of 10 years carrier of the p.Arg255* variant and the male precociously presented a very severe phenotype, while the 7-year-old PtA1 showed a moderate RTT picture (Table 1). In order to compare the cellular to the clinical phenotype, we measured morphological biomarkers in young neurons, one week from the progenitor stage and the electrophyiological functional markers at 120 days, when neurons should be definitely mature. Nuclear size reduction is reported both in animal models and in other iPSC-derived neurons: Yazdani et al. [36] observed in mouse embryonic stem cells that 3 days after plating WT and Mecp2^-/y^ lines had the same nuclear size, which in the WT neurons rapidly increased during the differentiation, while the defective lines remained significantly smaller. In our study, nuclear size appears to be a robust biomarker, early appreciable, with a significant difference only between WT and truncating variants. Neurons derived from clones expressing the mild variant p.Arg133Cys show a nuclear size similar to that of the WT isogenic clones and of neurons derived from healthy controls. Considering that the nuclear size correlates with transcriptional activity [36], it is meaningful that our mildest variant impacts scarcely or later on the *MECP2* transcriptional activity. Some studies discriminate between nuclear size and soma size [34], but according to our analyses the two parameters overlap. Chen at al [34] noticed a similar nuclear size in iPSC neurons expressing the p.Arg106Trp and isogenic WT lines, while it was consistently reduced in the neurons derived from knockout iPSCs. Should our clones with the truncating variants be assimilated to a KO system, the nuclear size could be considered a robust biomarker distinguishing MeCP2 missense from truncating variants. Similar to nuclear area size, we observed a significantly shortened neuronal length only in neurons of the case carrying the severe variant, consistent with the correlation of this biomarker with intellectual disability [37]. Interestingly these morphological parameters did not evidence heterogeneity across the isogenic WT neurons, suggesting that the individual variability does not seem to influence their values, while the comparison between WT isogenic neurons and neurons of healthy controls revealed a significant difference in the nuclear size. To interpret this discrepancy, we may hypothesize that the WT isogenic clones also expressed a minimum amount of the mutant MeCP2 protein, undetectable by Western blot, which influences the neuronal cells phenotype. As nuclear size is a reversible marker, it should be carefully followed-up when in vitro models are exposed to (epi) drugs. The study of end point and branch number referring to the dendritic harborization shows their homogeneous reduction in RTT neurons compared to healthy controls, raising the conclusion that whatever the deficit of MeCP2 it leads to impairment.

Moving to functional performance of RTT cells, the impairment of neuronal functionality was assessed at the late time point of 120 days [20] by testing the neurons’ capability to fire multiple action potentials (APs) as a sign of their maturation state. The data were compared to a pool of healthy wild type clones, and, when available, to isogenic controls for female clones. The male mutant cells were also compared to a male healthy control to take into consideration sex specific differences that were revealed by our study.

The distribution of cell firing patterns was unbalanced and indicates that maturation of the spiking pattern is strongly affected and widespread across the different MeCP2 mutations. Statistical analyses allowed the observation of a percentage of mature neurons decreasing from the girls with the p.Arg133Cys to those with the truncated variants reaching the minimum in the male case. Despite the heterogeneity reported in in vitro models, we incline to value the results obtained on the patients due to the lack of significant differences across all healthy control and WT isogenic clones.

Morphological data show that at 42 days, the variants resulting in more dramatic symptoms, p.Gly252Argfs*7 and p.Arg255*, already show reduced cellular complexity, while p.Arg133Cys does not display any difference. This differential phenotype is maintained at 120 days, as cell capacitance, a parameter determined by cell surface area, is dramatically decreased in the p.Gly252Argfs*7 and p.Arg255* neurons, whereas p.Arg133Cys neurons display, if any, an increased capacitance, suggesting that p.Arg133Cys differs for a qualitatively different effect on cell morphology.

Overall, our study on in vitro models from RTT cases, carriers of moderate and severe *MECP2* variants points to the reliability of specific morphological and functional markers, to distinguish across the phenotypic expression of the disease.

Interestingly, the concordance of precocious and mature assessments enforces the robustness of these models as well as the value of the identified biomarkers to measure the potential phenotypical rescue effected by novel drugs.

## 4. Materials and Methods

### 4.1. iPSC Generation and Characterization

Peripheral blood mononuclear cells (PBMCs) from RTT patients recruited for the study were collected and isolated with SEPMATE ^TM^ tubes (STEMCELL) according to the protocol. Erythroblast population was enriched by maintaining PBMCs for 10 days in SFEM Medium (STEMCELL) and then transducing with Cytotune 2.0 Sendai Reprogramming Kit (Life Technologies). After about 3 weeks the first iPSC colonies appeared on the mouse embryonic fibroblast (MEF) layer. Single clones were expanded and characterized for stemness and genomic stability.

Pluripotency marker expression was confirmed through immunofluorescence (using antibody against OCT3/4 and TRA-1-60) and RT-PCR (with primers for OCT3/4, SOX2 and NANOG). Karyotype analysis on at least 20 metaphases/sample allowed visualization of chromosome aberrations occurring during reprogramming. Refer to [38] for further details.

#### 4.1.1. SNP Array

Infinium HD Assay Ultra with Illumina multi-sample DNA Analysis BeadChips was performed to detect copy number variants CNVs (duplications, deletions, loss of heterozygosity) in iPSC clones compared with blood from the same case. The data were imported from iScan Control Software into GenomeStudio 2.0 Genotyping Module Software provided by Illumina for the analysis.

#### 4.1.2. Mutation Sequencing

In order to characterize iPSCs clones as wild-type (WT) or mutant (Mut), we checked the presence of the original MeCP2 mutation by comparing blood to iPSC cDNA. RNA was isolated from blood and iPSC clones with Quick-RNA^TM^ Mini Prep (Zymo Research). One μg was reverse-transcribed into cDNA by using SuperScript VILO cDNA Synthesis Kit (Thermo Fisher, Waltham, MA, USA). *MECP2* exon 4 was amplified with GoTaq DNA Polymerase (PROMEGA), using the following primers:Forward 5′ AAGCAAAGGAAATCTGGCCG 3′Reverse 5′ GTCTCCTGCACAGATCGGAT 3′

The obtained PCR fragment was purified with Illustra^TM^ ExoProStar^TM^ enzyme (GE Healthcare) and sequenced with the Sanger method, using the Big Dye Terminator (Applied Biosystems). The capillary electrophoresis was performed on the ABI PRISM 3500 Genetic Analyzer (Applied Biosystem), and output was analyzed with Sequencing Analysis Software 6 (Applied Biosystem).

#### 4.1.3. The Human Androgen Receptor Assay

To analyze X chromosome inactivation (XCI) patterns, the human androgen receptor (AR) assay was performed on genomic DNA extracted from blood with Wizard Genomic DNA Purification Kit (PROMEGA, Madison, WI, USA) according to manufacturer’s instructions. Then, 500 ng of genomic DNA was digested overnight at 37 °C with methylation-sensitive restriction enzyme *HhaI* (PROMEGA, Madison, WI, USA). During the digestion, the enzyme cuts only the unmethylated cytosines, preventing the subsequent amplification reaction. An amount of 250 ng of both digested and undigested DNA was amplified with primers designed on the polymorphic trinucleotide (CAG) repeats in the first exon of the human androgen receptor locus (HUMARA-AR): Forward 5′ GCTGTGAAGGTTGCTGTTCCTCAT 3′; Reverse 5′ TCCAGAATCTGTTCCAGAGCGTGC 3′. PCR was performed using a forward primer label with FAM fluorochrome on the 5′ end. PCR products were separated on an ABI PRISM 310 genetic analyzer (Applied Biosystem). The percentage of XCI was calculated as follows: (D1/U1)/(D1/U1 + D2/U2), where D1 and D2 are allele peak heights of the digested samples and U1 and U2 are the peaks of undigested genomic DNA. The XCI is considered balanced if the value obtained is between 15 and 85%, while it is considered unbalanced if >85% or <15%.

### 4.2. Cortical i-Neurons Generation

At least two characterized clones for each patient or control were differentiated into cortical i-neurons according to monolayer protocol, as described in [39].

### 4.3. Protein Extraction and Western Blot

Nuclear and cytoplasmic protein fractions were extracted and separated from 100 day i-neurons using NE-PER^TM^ Nuclear and Cytoplasmic Extraction Reagents (Thermo Fisher, Scientific, Waltham, MA, USA) supplemented with protease and phosphatase inhibitors (Roche) following the manufacturer’s instructions. Protein concentration was determined using DC Protein Assay (Biorad, Hercules, CA, USA). Nuclear extracts (15 μg) were separated on a 10% SDS-polyacrylamide gel electrophoresis and transferred to nitrocellulose (iBlot Gel Transfer Stacks Nitrocellulose, Thermo Fisher, Scientific Waltham, MA, USA). After blocking with 5% non-fat dry milk, TS1X Buffer (20mM Tris-HCl pH 7.5, 150 mM NaCl) and 0.3% Tween-20 for 1 h, membranes were incubated with primary antibodies (Anti-MeCP2 C-ter (1:500) ab2829, Abcam; Anti-MeCP2 N-ter (1:500) M7443, Sigma-Aldrich; Anti-Histone H3 (1:2000) ab1791, Abcam) overnight at 4 °C, and then with secondary antibodies (goat Anti-Mouse HRP AP124P (1:3000), goat Anti-rabbit HRP AP307P (1:3000), both from Millipore) for 2 h at RT. The Clarity^TM^ Western ECL Substrate (Biorad, Hercules, CA, USA) was used for detection of the HRP-conjugated secondary antibody.

### 4.4. Immunofluorescence Staining

i-Neurons were fixed in 4% paraformaldehyde (20 min, 37 °C) at 42 days for morphological analysis and 100 days for MeCP2 localization. Primary antibodies in gelatin dilution buffer (0.2% gelatin, 0.3% Triton-X-100, 20 mM sodium phosphate buffer pH 7.4, 0.45 M NaCl, all by Sigma) were incubated overnight at 4 °C. Anti-MAP2 (1:200, ab32454 Abcam) and Anti-TUJ1 (1:200, MMS-435P, Covance) were used to identify neuronal cells, while Anti-MeCP2 C-ter (1:100, ab2829, Abcam) and Anti MeCP2 N-ter (1:40, ab2828, Abcam) were used to discriminate different portions of MeCP2 protein. Secondary antibodies Goat Anti-Mouse-IgG (H + L) 488 (A-11001, Invitrogen) and F (ab’)2-Goat Anti-Rabbit-IgG (H + L) 555 (A-21430, Invitrogen) were diluted 1:300 and incubated for 2 h at RT. Nuclei were stained with DAPI, 1024 × 1024 pixel images were acquired with a Nikon Eclipse Ti microscope, 40× objective was used for arborization and nuclear area measurement and 60× objective for MeCP2 localization. Scale bars are reported in all figures.

### 4.5. Morphological Analysis

Experiments to outline the morphology of cells were performed on 42 day-old neurons, stained with antibodies against MAP2 and TUJ1 neuronal markers. The Skeletonize Image J Plugin was used for the analysis of at least 50 cells selected for each sample and three independent experimental repeats. Refer to [40] for further details.

### 4.6. Electrophysiological Recordings

When the i-neurons reached 120 days of differentiation electrophysiological experiments were performed. Cell plates from the incubator were directly put under the microscope an Axioskop (Zeiss, Oberkochen, Germany) equipped with 60× lens. During the experiment the neural differentiation medium (NDM) was replaced by artificial cerebrospinal fluid (ACSF: in mM NaCl 119, KCl 2.5, NaHPO_4_ 1.25, NaHCO_3_ 15, HEPES 10, glucose 12.5, CaCl_2_·4H_2_O_2_, MgSO_4_·7H_2_O_2_; pH = 7.3 ± 0.1; osmolarity: 295 ± 5 mOsm) which was perfused continually at 1 ml/min at 35 °C. For the recording, borosilicate pipettes with internal and external diameters respectively 0.86 mm and 1.5 mm (WPI) were pulled using a P-97 (Sutter instruments) to reach resistance of 3–5 MΩ when filled by internal solution (in mM: K-Gluconate 130, HEPES 10, EGTA 1, CaCl_2_ 0.3, MgCl_2_ 1, ATP 4, GTP 0.3, phosphocreatine 5; pH = 7.3 ± 0.1; osmolarity: 285 ± 5 mOsm). For experiments on PtB1 (p.Arg133Cys) and isogenic WT we used the same external solution but a high chloride internal solution (in mM: KCl 120, K-gluconate 10, HEPES 10, EGTA 1, CaCl_2_ 0.3, MgCl_2_ 1, ATP 4, GTP 0.3, phosphocreatine 5; pH = 7.3 ± 0.1; osmolarity: 285 ± 5 mOsm). During the recordings, somas of the neurons were reached by the tip of the recording pipette and a positive pressure was applied. Quickly releasing the pressure allowed the formation of the giga-seal after which a gentle positive pressure was applied to break the membrane achieving the whole seal configuration. Recordings were made in voltage-clamp and current-clamp mode holding potential of −75 mV using a Multiclamp 700 A amplifier controlled from a PC by Clampex 9.2 via a Digidata 1322 A (Molecular Devices). LJP was calculated a priori, and the recording adapted to reach wanted values. Data were low-pass filtered at 1 kHz and sampled at 10 kHz.

### 4.7. Statistical Analysis

Statistical analysis was performed using Graph Pad Prism 7 program and Glabstab tool. One-way ANOVA and Kruskal–Wallis post hoc test were chosen for multiple comparisons. All data were expressed as mean +/− SEM. Data were obtained from three independent experiments. Morphological alterations and nuclear size were evaluated by selecting at least 50–80 neurons for each sample. E-recordings were evaluated on at least 20–30 cells for a sample.

## Figures and Tables

**Figure 1 ijms-23-14491-f001:**
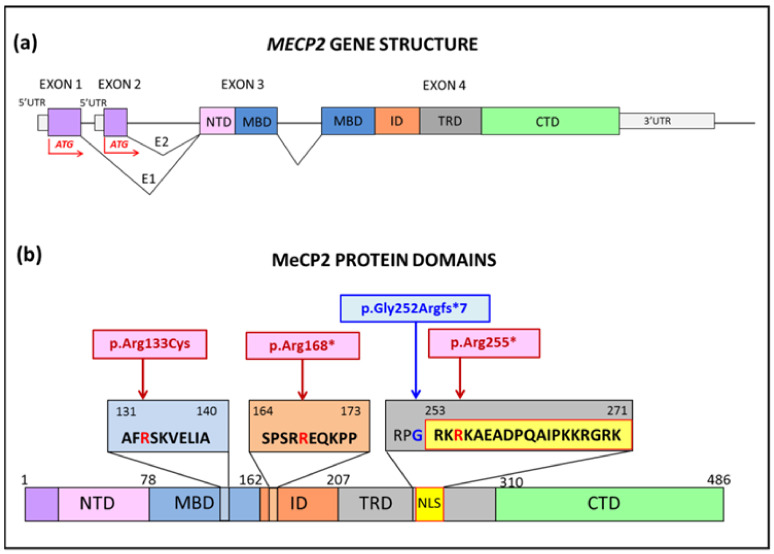
Representation of *MECP2* gene structure and protein domains. (**a**) Schematic of *MECP2* gene organization with four exons and the two isoforms (E1 and E2). (**b**) Impact of the selected hot-spot pathogenic variants on MeCP2-E2 protein isoform. NTD: N-Terminal Domain, MBD: Methyl CpG Binding Domain, ID: Intervening Domain, TRD: Transcriptional Repressor Domain, NLS: Nuclear Localization Signal, CTD: C-Terminal Domain.

**Figure 2 ijms-23-14491-f002:**
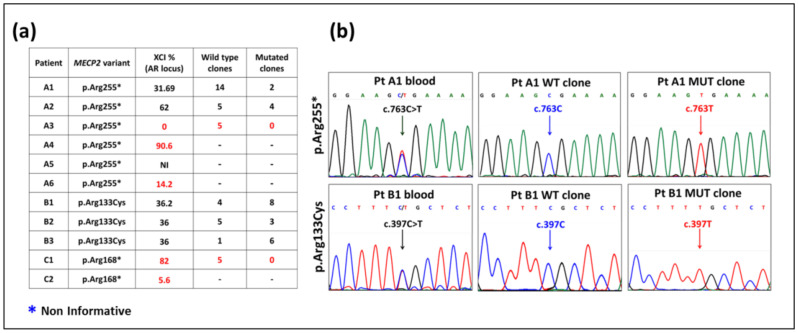
XCI pattern and selective expression of wild-type (WT) and pathogenic variant (MUT) allele in RTT female iPSC lines. (**a**) The table sums up XCI analyses. Patients with XCI > 80–85% or <15–20% were considered unbalanced and are highlighted by red characters. In patient A5, non-informative for the AR locus (*), XCI was investigated at DXS6673E locus (XCI = 13.5%). The number of wild-type and mutant clones obtained are reported respectively in the fourth and fifth column of the table. (**b**) Electropherograms of cDNA sequences from p.Arg255* and p.Arg133Cys RTT girls displaying biallelic expression of *MECP2* allele in the blood and monoallelic expression of WT or MUT allele in the iPSC isogenic clones.

**Figure 3 ijms-23-14491-f003:**
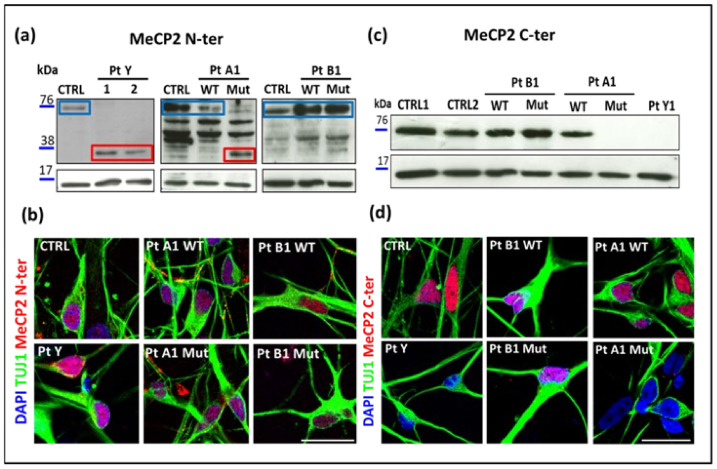
MeCP2 protein in i-neurons from RTT cases with truncating and missense variants. (**a**,**c**) Western blot on protein extracted from neurons of controls (CTRL), patients carriers of p.Gly252Argfs*7 (PtY), p.Arg255* (PtA1), p.Arg133Cys (PtB1), wild type (WT) and mutant (Mut) isogenic clones, using N-ter antibody (Ab) (**a**) and C-ter Ab (**c**). The blue rectangles frame the band representative of the full-length protein (~75 kDa), while the red ones frame the 37 kDa truncated isoform. Histone H3 (15 kDa) was used as loading control. Panels (**b**) and (**d**) show immunostaining of mature neurons with beta 3 tubulin (TUJ1), highlighting the localization of MeCP2 (red). Nuclei were counterstained with DAPI; scale bar: 20 μm.

**Figure 4 ijms-23-14491-f004:**
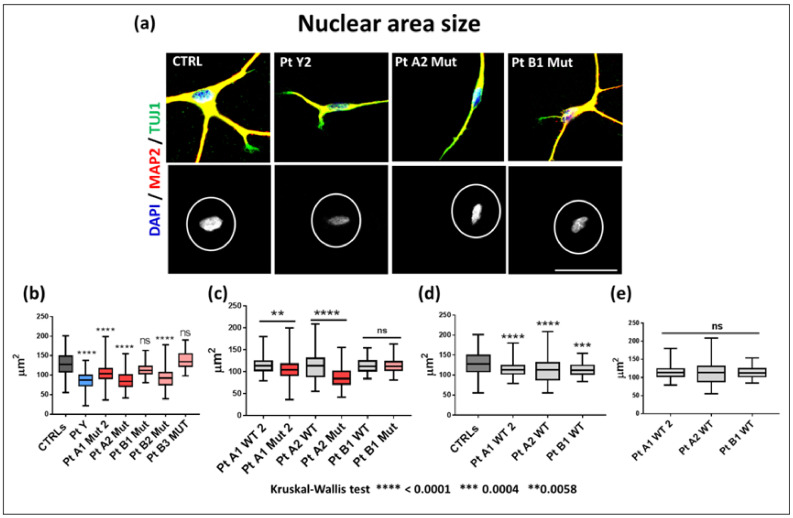
Nuclear area size of young RTT neurons. (**a**) IF staining with pan-neuronal markers MAP2 (red) and TUJ1 (green), counterstaining with DAPI and analysis with Image J software of single young neurons (top panels) and scanned nuclear area size (bottom panels) of a control, PtY2 (p.Gly252Argfs*7), PtA2 Mut and PtB1 Mut i-Neurons. Scale bar: 50 μm. (**b**) Box-plots representative of the comparison for nuclear area of each mutant clone and a pool of eight healthy controls, four males and four females (CTRLs), (**c**) each isogenic WT clone vs. the correspondent mutant carrying the p.Arg255* and p.Arg133Cys, (**d**) between each isogenic WT clone and a pool of controls, (**e**) across WT isogenic clones. Data were compared using Kruskal-Wallis test. **** *p*-value < 0.0001, *** *p*-value = 0.0004, ** *p*-value = 0.0058. ns: not significant.

**Figure 5 ijms-23-14491-f005:**
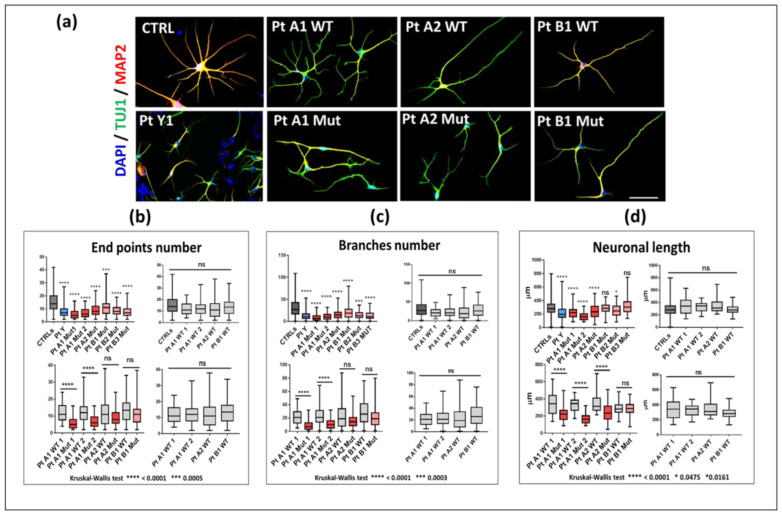
Morphological analysis of young RTT neurons. (**a**) IF of MAP2 (red), TUJ1 (green), and DAPI (blue). Scale bar: 50 μm. (**b**–**d**) Box-plots from the comparison between mutant (red) versus isogenic WT clones (light grey) of RTT females and between the male (blue) versus controls (dark grey) for end points number, branches number and neuronal length. Decreased number of end points (**b**), branches (**c**), and neuronal length (**d**) were highly significant in the Kruskal–Wallis test, while no significance was proved in the comparison among wild-type clones. Data were compared using Kruskal-Wallis test. In (**b**) **** *p*-value < 0.0001, *** *p*-value = 0.0005; in (**c**) **** *p*-value < 0.0001, *** *p*-value = 0.0003; in (**d**) **** *p*-value < 0.0001, * *p*-value = 0.0475 and 0.0161. ns: not significant. A correction was also introduced by Bonferroni test.

**Figure 6 ijms-23-14491-f006:**
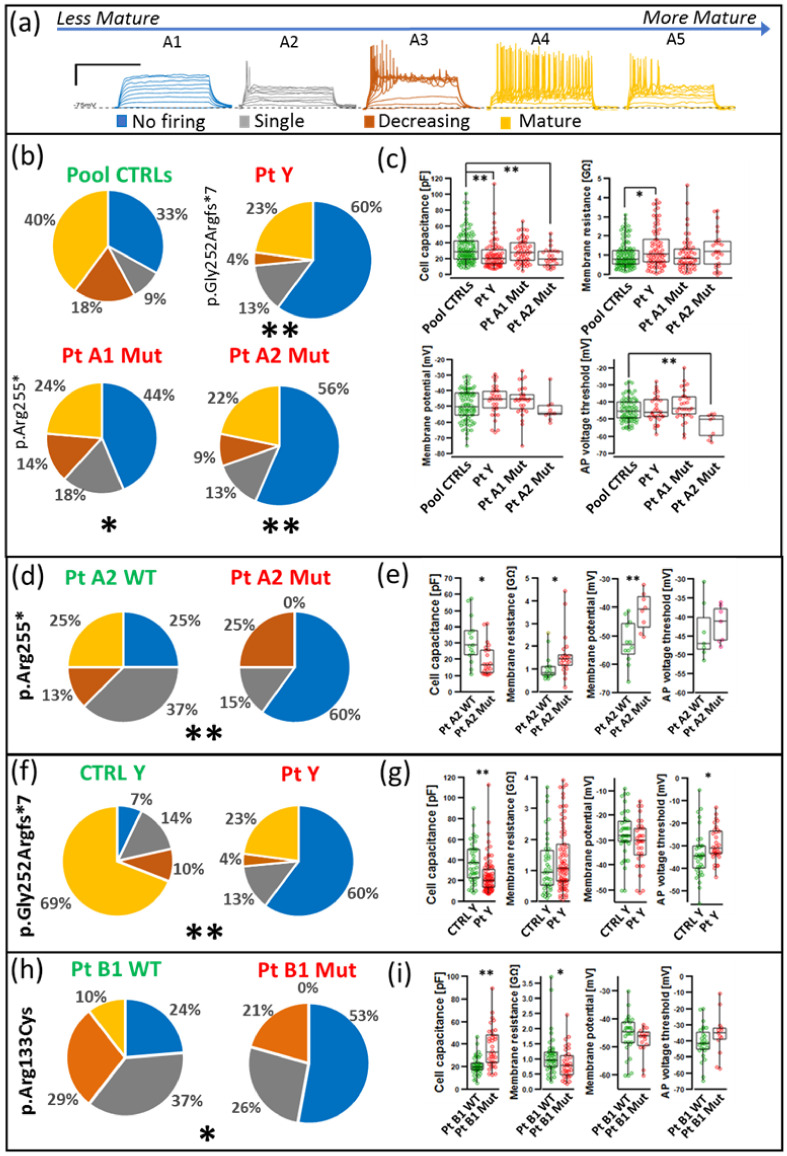
Differential alteration of firing and biophysical properties in neurons carrying different *MECP2* mutations. (**a**) representative traces of four grades of firing classes using a code color from no firing (A1 blue) to single firing (A2 grey), decreasing firing (A3 orange) until increasingly mature neurons A4 and A5 (yellow) (calibration 1 s, 30 mV). (**b**,**d**,**f**,**h**): distribution according to the color code of the four firing classes in the neuronal populations from controls and RTT cases. The chi-square test was used to compare (**b**) mutated group versus controls pool (PtY *p* < 0.01, PtA1 Mut *p* = 0.02, PtA2 Mut *p* < 0.01), (**d**) the PtA2 mutated group versus its WT isogenic control (*p* < 0.01), (**f**) the PtY mutated group versus the male control group (*p* < 0.01), and (**h**) the PtB1 mutated group versus its isogenic WT control (*p* < 0.01); (**c**,**e**,**g**,**i**): cell capacitance, membrane resistance, membrane potential, and action potential voltage threshold for the clones shown in (**b**,**d**,**f**,**h**). Cell capacitance, membrane resistance, membrane potential, and action potential voltage threshold data were not parametric and are reported as scatter plot on box-plots in (**c**,**e**,**g**,**i**) representing 90, 75, 50, 25, and 10% of the distribution. Data were compared using Mann–Whitney test. * *p*-value from 0.05 to 0.01, ** for *p*-value < 0.01.

**Table 1 ijms-23-14491-t001:** Clinical features of patients included in the study.

Pt	Age	Age at Diagnosis	Clinical Signs at Onset	Epilepsy Onset	D.R.	Motor Function	Speech	Communication	Behavioural Problems	S	GI Problems	GF	M	CSS
A1	7 years	3 years	Regression, hand stereotypies (24 months)	5 years	Yes	Independent walking	Absent	Maintained through eye contact	Absent	No	No	Yes	Yes	20
A2	10 years	1 year	Developmental delay, hand stereotypies (12 months)	24 months	Yes	Never acquired deambulation	Absent	Absent	Yes	Yes	Mild	No	Yes	33
B1	19 years	5 years	Regression, absent speech, hand stereotypies (30 months)	24 months	No	Independent walking	Single words	Maintained through eye contact	Absent	Mild	Mild	Yes	Yes	17
B2	45 years	30 years	Regression, autistic features, hand stereotypies (18 months)	42 months	No	Independent walking	Absent	Maintained through eye contact	Absent	Mild	Mild	No	NA	18
B3	22 years	3 years	Regression, hand stereotypies (18 months)	18 months	No	Independent walking	Absent	Maintained through eye contact	Absent	Mild	No	Yes	Yes	17
Y	dead	11 months	Decreased fetal movements and fetal growth restriction, oculogyric crisis and generalized hypotonia (prenatal-neonatal)	-	No	NA	NA	NA	NA	NA	Yes (constipation)	Yes	Yes	

DR: Drug Resistance, S: Scoliosis, GI: Gastrointestinal, GF: Growth Failure, M: Microcephaly, CSS: Clinical Severity Scale, NA: Not Applicable.

## Data Availability

The data presented in this study will be available in a dedicated repository, as soon as the manuscript is accepted.

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
