# Peer review of "Modeling RTT Syndrome by iPSC-Derived Neurons from Male and Female Patients with Heterogeneously Severe Hot-Spot MECP2 Variants"

_ijms, 2022, doi:10.3390/ijms232214491_

Round 1
Reviewer 1 Report
The manuscript by Perego et al., entitled ‘Modeling RTT syndrome by ”in vitro” i-Neurons from male 2 and female patients with heterogeneously severe hot-spot 3 MECP2 variants' describes the generation and the morphological and functional characterization of new in vitro iPSC-neurons from blood of RTT patients carrying p.Arg133Cys p.Arg255* (female patients),and p.Gly252Argfs*7 (male patient) variants, responsible for phenotypes with a different severity.
Overall, this is a well-written manuscript. The authors have designed the right experiments to answer certain questions thoroughly. I expect that this manuscript will be of interest for the readership of Rett syndrome research.
I have minor comments and suggestions that I would like the authors to address prior to recommending this manuscript for publication, as follows.
Lines 5-6: The authors should carefully check the affiliations and their order. For example, the affiliation #7, referred to Lidia Larizza, is missing.
Lines 89-91.
In the sentence "we compared iPSCs from a male with Gly252Argfs*7, two girls with the nearby Arg255* hot-spot variant and a female with the Arg133Cys variant, associated with mild clinical manifestations", the authors should better specify that both Gly252Argfs*7 and Arg255* are associated with a more severe phenotype as compared with the Arg133Cys variant. This phrase might be changes as follows: "we compared iPSCs from a male with Gly252Argfs*7, two girls with the nearby Arg255* hot-spot variant and a female with the Arg133Cys variant, this latter associated with mild clinical manifestations"
Lines 89-91.
The sentence "We generated iPSCs from Peripheral Blood Mononuclear Cells (PBMC) of MECP2 defective girls, obtaining isogenic clones, expressing either the WT or the Mut allele, then differentiated into prefrontal cortical neurons and compared to healthy controls to appoint morphological biomarkers in young neurons and electrophysiological biomarkers in mature neurons" is too long and twisted. Please, revise this sentence.
Lines 131-137.
The text "cDNA sequencing to disclose whether each clone expressed 131 either WT/MUT or both alleles, confirmed the occurrence of isogenic clones expressing the variant 132 or wild-type allele in the girls with balanced XCI (Figure 2b), while only the wild-type allele was 133 expressed in the skewed cases. As regards the RTT male carrier of p.Gly252Argfs*7, two clones (PtY1 134 and PtY2), out of several obtained by reprogramming peripheral blood cells to iPSCs, were selected 135 for this study. In parallel to RTT control iPSCs clones were obtained and characterized from 4 136 healthy males and 4 healthy females (CTRLs)" appears included in the Figure 2 legend. I suppose that it should be moved in the main text of the manuscript.
Moreover, in the last sentence of the text, "In parallel to RTT control iPSCs clones were obtained and characterized from 4 healthy males and 4 healthy females (CTRLs)", I suggest to add "," after "In parallel to RTT control".
Table 1.
Please, format the table using the same font and size.
Line 157.
Please, correct " DR: Drug ResistaDR: Drug Resistance"
Supplementary Figure S1 Legend.
The description of the (c) is long and quite confusing. Please, revise.
Lines 169-170.
Please, specify the meaning of i-Neurons (does it mean induced-Neurons?).
Lines 173-174.
The authors should describe the number of clones analyzed for the male patient Y and for the female patient A1.
Please, specify how many clones you analyzed for patients A2, B1, B2 and B3.
Supplementary Figure S1 and section "2.3. From iPSCs to Neuron Differentiation" of the main text (lines 160-174)
I suppose that the authors performed immunostaining of the pluripotency markers OCT3/4 and Tra-1-60, RT-PCR of OCT3/4, SOX2, and NANOG, and karyotyping in clones from all patients and controls. Although it is reasonable to show data obtained only in one clone per patient, the authors should include in the Supplementary Figure S1 a, b and c also data obtained in cells from patients A2, B2 and B3. Moreover, the authors should specify in Fig. S1a whether IF refers to Mut or WT clone, as illustrated in Fig. S1c.
The authors might write in the main text or in the figure legend that these data are representative of all clones analyzed.
Lines 176-178.
As suggested in the previous comment, the authors should insert in the Figure 3a and c also WB data obtained in patients A2, B2 and B3. Alternatively, they should write in the text that A2, B2 and B3 clones displayed similar results.
Lines 177.
Taking into account that MeCP2 A (E2) and B (E1) isoforms differ at their N-terminal region, is the N-Ter MeCP2 antibody designed to recognize both isoforms or it binds specifically only one of them? The authors should specify this point in the main text.
Lines 26, 194 and 425.
Please, change "Immunofluorescence" with "immunofluorescence" (all in lower case).
Line 201.
I suggest to insert a schematic representation of the neuronal differentiation, in which the authors show the different temporal stages, in days of differentiation (eg: progenitors, mature neurons etc.). This scheme may help the reader in understanding the results. For instance, in line 201, it is not clear whether "42 days of neuronal differentiation" is the stage of progenitor or the experimental point.
Figure 4 and lines 203-204.
In lines 203-204 the authors wrote "patients from the mutated clones of all RTT girls". Considering that different patients carrying the same mutation could display a different morphological phenotype, I suggest to include in the Fig. 4 one image per patient, including also A1, B2 and B3.
Line 205.
The authors should specify which controls are included in the "pool of 42 days controls’ neurons" (both healthy and isogenic patient-derived controls?). Moreover, in the Figure 4a legend, control cells are indicated generically as "a control". Please, use a similar description in the main text and figure legend.
Figure 4b.
What do you mean with "Pt A1 Mut 2"? Is it the clone #2 of the A1 patient?
Line 212.
In "CTRLs on Yax", do you mean the X axis?
Lines 220-222.
The authors showed that the nuclear size is statistically similar among female isogenic controls (Fig 4e), whereas each isogenic control is statistically different from the pool of controls (Fig. 4D). I understand the lack of isogenic control for the male patient, but for female patients it could be appropriate to show only data compared to their isogenic control (Fig. 4c), including also B2 and B3 patients.
Figure 5a.
I suggest to include also representative IF images from B2 and B3 patients and from isogenic controls of A1, A2 and B1.
Lines 250-251.
The authors tested the maturity of mutant and wild-type cells at functional level by electrophysiological assays. I suggest to describe in the main text the stage of neuronal differentiation in which they performed these analyses, as they done for the morphological assays. I noticed that they discussed the different time points analyzed in morphological and electrophysiological assays in the "Discussion" section, but I suppose that it could be useful for the reader to find this detail also in the "Results" section.
Line 260.
Does the pool of control cells include only healthy controls or even isogenic controls? The authors should specify this at least in the legend of Fig. 6 and Supplementary Fig. 3.
Fig. 6.
Please, use for this figure the same font and size used for the other figures.
Lines 268-271.
The authors say that the p.Arg255* variant showed altered cell capacitance, membrane resistance, resting potential and the action potential threshold only in one patient and they suggested a possible involvement of genetic background. However, they did not show the statistical significance for the p.Arg255* variant in the graphs depicting membrane resistance and membrane potential. Are these data statistically significant?
Lines 293-294.
The distribution of firing classes relative to the CTRL Y is strongly different from that of the Pool CTRLs, likely also considering the gender difference. The authors might discuss this evidence in the "Discussion" section.
Lines 271-302.
The order of description of results obtained in the different female patients is quite confusing, also considering that Arg255* data are compared with both the Pool CTRLs and with its isogenic control, while Arg133Cys data are compared only with its isogenic control. Also, the description of results reported in 6h and 6i in two different subsections of the paragraph is quite confusing. The authors might slightly reorganize the description of this paragraph, also grouping data from Arg133Cys patient.
Lines 303-306.
The authors should write in the main text that, besides the firing frequency (Supplementary Fig 3c), also the other properties are unchanged between PtY1 and PtY2 (Supplementary Fig 3d).
Lines 310-312.
Why the authors compare the Arg255* mutant to the pooled CTRL cells, while the Arg133Cys was compared to its isogenic control?
Lines 386-388.
This sentence is confusing and misleading. Please, clarify better this point.
Discussion section.
As reported in the previous comments, the authors could add a small part in which they discuss the use of pooled controls (all healthy controls or healthy + isogenic controls?) and isogenic controls in the experiments performed in mutant females.
Author Response
Comments and Suggestions for Authors
The manuscript by Perego et al., entitled ‘Modeling RTT syndrome by ”in vitro” i-Neurons from male 2 and female patients with heterogeneously severe hot-spot 3 MECP2 variants' describes the generation and the morphological and functional characterization of new in vitro iPSC-neurons from blood of RTT patients carrying p.Arg133Cys p.Arg255* (female patients),and p.Gly252Argfs*7 (male patient) variants, responsible for phenotypes with a different severity.
Overall, this is a well-written manuscript. The authors have designed the right experiments to answer certain questions thoroughly. I expect that this manuscript will be of interest for the readership of Rett syndrome research. We thank the reviewer for appreciating our work and its potential interest for the readership of Rett syndrome research
I have minor comments and suggestions that I would like the authors to address prior to recommending this manuscript for publication, as follows.
Lines 5-6: The authors should carefully check the affiliations and their order. For example, the affiliation #7, referred to Lidia Larizza, is missing. ok
Lines 89-91.
In the sentence "we compared iPSCs from a male with Gly252Argfs*7, two girls with the nearby Arg255* hot-spot variant and a female with the Arg133Cys variant, associated with mild clinical manifestations", the authors should better specify that both Gly252Argfs*7 and Arg255* are associated with a more severe phenotype as compared with the Arg133Cys variant. This phrase might be changes as follows: "we compared iPSCs from a male with Gly252Argfs*7, two girls with the nearby Arg255* hot-spot variant and a female with the Arg133Cys variant, this latter associated with mild clinical manifestations"
We thank the reviewer for the suggested sentence which has replaced the previous incomplete one
Lines 89-91.
The sentence "We generated iPSCs from Peripheral Blood Mononuclear Cells (PBMC) of MECP2 defective girls, obtaining isogenic clones, expressing either the WT or the Mut allele, then differentiated into prefrontal cortical neurons and compared to healthy controls to appoint morphological biomarkers in young neurons and electrophysiological biomarkers in mature neurons" is too long and twisted. Please, revise this sentence.
We split the tortuous sentence in two hopefully more readable sentences
Lines 131-137.
The text from line 131 to 136 "cDNA sequencing to disclose whether each clone expressed either WT/MUT or both alleles, confirmed the occurrence of isogenic clones expressing the variant or wild-type allele in the girls with balanced XCI (Figure 2b), while only the wild-type allele was expressed in the skewed cases. As regards the RTT male carrier of p.Gly252Argfs*7, two clones (PtY1 134 and PtY2), out of several obtained by reprogramming peripheral blood cells to iPSCs, were selected for this study. In parallel to RTT control iPSCs clones were obtained and characterized from 4 healthy males and 4 healthy females (CTRLs)" appears included in the Figure 2 legend. I suppose that it should be moved in the main text of the manuscript.
Moreover, in the last sentence of the text, "In parallel to RTT control iPSCs clones were obtained and characterized from 4 healthy males and 4 healthy females (CTRLs)", I suggest to add "," after "In parallel to RTT control".
We added the indicated paragraph to the main text
Table 1.
Please, format the table using the same font and size.
We used the consented format.
Line 157.
Please, correct " DR: Drug ResistaDR: Drug Resistance"
done
Supplementary Figure S1 Legend.
The description of the (c) is long and quite confusing. Please, revise.
We shortened point (c) of the legend
Lines 169-170.
Please, specify the meaning of i-Neurons (does it mean induced-Neurons?).
done
Lines 173-174.
The authors should describe the number of clones analyzed for the male patient Y and for the female patient A1.
Please, specify how many clones you analyzed for patients A2, B1, B2 and B3.
We rephrased the text to clarify the point requested by the reviewer
Supplementary Figure S1 and section "2.3. From iPSCs to Neuron Differentiation" of the main text (lines 160-174)
I suppose that the authors performed immunostaining of the pluripotency markers OCT3/4 and Tra-1-60, RT-PCR of OCT3/4, SOX2, and NANOG, and karyotyping in clones from all patients and controls. Although it is reasonable to show data obtained only in one clone per patient, the authors should include in the Supplementary Figure S1 a, b and c also data obtained in cells from patients A2, B2 and B3. Moreover, the authors should specify in Fig. S1a whether IF refers to Mut or WT clone, as illustrated in Fig. S1c.
We confirm that the indicated characterizations were performed in clones from all patients and controls
The authors might write in the main text or in the figure legend that these data are representative of all clones analyzed.
We added the proper sentence suggested by the referee
Lines 176-178.
As suggested in the previous comment, the authors should insert in the Figure 3a and c also WB data obtained in patients A2, B2 and B3. Alternatively, they should write in the text that A2, B2 and B3 clones displayed similar results.
We thank the reviewer for the suggestion, but we could not perform the WB of the PtA2 Mut and Wt after hybridization with the N-ter Ab, due to the low amount of the obtained proteins. The WB with the C-ter Ab confirmed the absence of the fragment corresponding to the wild-type MeCp2 in PtA2 Mut and its presence in PtA2WT. These WB will be uploaded in the repository being thus available to the reviewer’s check .
As regards the missense p.Arg133Cys, we analyzed only one patient as a regular MeCP2 band is expected for both wild-type and Mut clones. This is the reason why we did not consider to replicate the experiments in the other cases.
Line 177.
Taking into account that MeCP2 A (E2) and B (E1) isoforms differ at their N-terminal region, is the N-Ter MeCP2 antibody designed to recognize both isoforms or it binds specifically only one of them? The authors should specify this point in the main text.
We verified that our antibody recognizes both the isoforms, because it links to the aminoacidic sequence
DQDLQGLKDKPLKFKK-C.
Lines 26, 194 and 425.
Please, change "Immunofluorescence" with "immunofluorescence" (all in lower case).
Done
Line 201.
I suggest to insert a schematic representation of the neuronal differentiation, in which the authors show the different temporal stages, in days of differentiation (eg: progenitors, mature neurons etc.). This scheme may help the reader in understanding the results. For instance, in line 201, it is not clear whether "42 days of neuronal differentiation" is the stage of progenitor or the experimental point.
Thanks for your suggestion: we modified the sentence in order to distinguish between progenitors and young neurons time-points. The entire workflow is reported in the cited manuscripts.
Figure 4 and lines 203-204.
In lines 203-204 the authors wrote "patients from the mutated clones of all RTT girls". Considering that different patients carrying the same mutation could display a different morphological phenotype, I suggest to include in the Fig. 4 one image per patient, including also A1, B2 and B3.
We chose to show images representative of each MECP2 variant described in this paper, not to overload the manuscript. The statistical results, based on the analysis of more than 50-80 cells for each clonal culture, have been reported for all the patients (figs 4 and 5). The pictures from all the patients will be uploaded in the repository.
Line 205.
The authors should specify which controls are included in the "pool of 42 days controls’ neurons" (both healthy and isogenic patient-derived controls?). Moreover, in the Figure 4a legend, control cells are indicated generically as "a control". Please, use a similar description in the main text and figure legend.
We thank the reviewer for noticing the non exact comment to our image. The represented control’s neuron is a neuron obtained from a healthy control. CTRLs indicate only healthy controls, while isogenic are reported as WT.
Figure 4b.
What do you mean with "Pt A1 Mut 2"? Is it the clone #2 of the A1 patient? yes
Line 212.
In "CTRLs on Yax", do you mean the X axis?
We amended the mistake
Lines 220-222.
The authors showed that the nuclear size is statistically similar among female isogenic controls (Fig 4e), whereas each isogenic control is statistically different from the pool of controls (Fig. 4D). I understand the lack of isogenic control for the male patient, but for female patients it could be appropriate to show only data compared to their isogenic control (Fig. 4c), including also B2 and B3 patients.
We agree with the reviewer, but unfortunately we could not obtain from B2 and B3 isogenic clones enough cells to do a statistical analysis
Figure 5a.
I suggest to include also representative IF images from B2 and B3 patients and from isogenic controls of A1, A2 and B1.
Indeed, isogenic controls of A1, A2 and B1 are represented in Fig 5a. For B2 and B3 we could not show the images for the reason explicited in the previous comment.
Lines 250-251.
The authors tested the maturity of mutant and wild-type cells at functional level by electrophysiological assays. I suggest to describe in the main text the stage of neuronal differentiation in which they performed these analyses, as they done for the morphological assays. I noticed that they discussed the different time points analyzed in morphological and electrophysiological assays in the "Discussion" section, but I suppose that it could be useful for the reader to find this detail also in the "Results" section.
We introduced in the “Results” Section the differentiation stage of the neurons used for e-recordings.
Line 260.
Does the pool of control cells include only healthy controls or even isogenic controls? The authors should specify this at least in the legend of Fig. 6 and Supplementary Fig. 3.
We specified in the text (p. 9) that the controls do not include the isogenic controls
Lines 268-271.
The authors say that the p.Arg255* variant showed altered cell capacitance, membrane resistance, resting potential and the action potential threshold only in one patient and they suggested a possible involvement of genetic background. However, they did not show the statistical significance for the p.Arg255* variant in the graphs depicting membrane resistance and membrane potential. Are these data statistically significant?
We are sorry for the confusion created by our wording. The referee is correct in saying that there is no difference in membrane resistance and membrane potential in fig. 6c, however the alteration of these parameters emerges from the comparison with the isogenic clone in fig. 6e. We added to the text the reference to this figure.
Lines 293-294.
The distribution of firing classes relative to the CTRL Y is strongly different from that of the Pool CTRLs, likely also considering the gender difference. The authors might discuss this evidence in the "Discussion" section.
Thank you for pointing this out. We added a sentence in the discussion “The data were compared to a pool of healthy wild type clones, and, when available, to isogenic controls for female clones. The male mutant cells were also compared to a male healthy control to take into consideration sex specific differences that were revealed by our study”
Lines 271-302.
The order of description of results obtained in the different female patients is quite confusing, also considering that Arg255* data are compared with both the Pool CTRLs and with its isogenic control, while Arg133Cys data are compared only with its isogenic control. Also, the description of results reported in 6h and 6i in two different subsections of the paragraph is quite confusing. The authors might slightly reorganize the description of this paragraph, also grouping data from Arg133Cys patient.
We rephrased the paragraph description of the results shown in figs. 6h and 6i.
Lines 303-306.
The authors should write in the main text that, besides the firing frequency (Supplementary Fig 3c), also the other properties are unchanged between PtY1 and PtY2 (Supplementary Fig 3d).
We added this specifications in the Results.
Lines 310-312.
Why the authors compare the Arg255* mutant to the pooled CTRL cells, while the Arg133Cys was compared to its isogenic control?
The analysis of the Arg255* mutant and the isogenic control cells was performed in a set of experiments that was designed to study also inhibitory synaptic currents. For this reason, the solutions used were slightly different from those used in the other experiments including the data of the pooled controls. Therefore, we decided to skip the comparison between pooled controls and the Arg255* mutant because of the non-homogenous recording condition.
Lines 386-388.
This sentence is confusing and misleading. Please, clarify better this point.
We rephrased the concept and hope now it is clearer
Discussion section.
As reported in the previous comments, the authors could add a small part in which they discuss the use of pooled controls (all healthy controls or healthy + isogenic controls?) and isogenic controls in the experiments performed in mutant females.
We added to the Discussion the sentence: “The data were compared to a pool of healthy wild type clones, and, when available, to isogenic controls for female clones. The male mutant cells were also compared to a male healthy control to reveal possible sex specific effects.”
Reviewer 2 Report
Summary
The authors generated iPSCs and iPSC-derived neurons from the PBMC of multiple Rett Syndrome patients containing different MECP2 mutations, and comprehensively characterized the morphological defects and electrophysiological abnormalities of RTT neurons, showing that the strength of cellular phenotypes correlates with the clinical severity of the human patients.
This work has two major contributions:
1. One is the systematic generation of a collection of iPSC-derived RTT neurons from both male and female patients, with 3 different genetic mutations causing either severe or mild clinical symptoms. The authors also generated isogenic controls by leveraging balanced XCI cell lines, making the phenotype evaluation more precise. This set of cell lines can be valuable for future research on RTT and MeCP2.
2. The second contribution is the evaluation of cellular phenotypes in iPSC-derived RTT neurons, including morphological measurements and electrophysiological analysis. The phenotypes of nuclear size, branch number, and immature electrical properties are largely expected from these neurons based on previous findings, but it is great that the authors establish a correlation between phenotype strength and clinical severity of the patients, providing a useful biomarker moving forward.
Overall, this work is worth publishing given the contributions of creating multiple new iPSC lines for studying 3 different genetic mutations in RTT neurons, and the phenotypic validation of these lines. Before publishing, I'd like the authors to address the comments below (some are marked optional).
General Comments
1. [optional improvement] The main limitation of this work are: (1) the lack of new scientific discoveries or mechanistic insights, and (2) the authors did not show evidence that these iPSC-derived neurons can be used for effective/reliable drug testing (e.g. if we can see robust results in these neurons following drug treatments). More data on either one of these directions can significantly improve the strength of this manuscript. Given the strengths of this manuscript, improving upon these limitations is optional.
2. The authors often use the word "altered" to describe the changes in electrophysiological properties (e.g. cell capacitance, membrane resistance). When possible (i.e. the referred changes are in one direction), these descriptions should be made more precise by saying whether it's increased or decreased.
3. In Figure 6, Pt A2's mutant strain was used in two analysis (Fig 6b/c and Fig 6d/e) but the results are quite different. In 6c, the Pt A2 Mut has reduced membrane potential and AP threshold compared to control, but in 6e, both phenotypes are in the increasing direction. While variability between different sets of trials is understandable, I'd expect the change to be at least in the same direction. Having them in different directions suggests there are a lot of variabilities in the experimental protocol.
Specific Comments
1. The word "i-Neurons" in the paper title is confusing, and not a common way to refer to iPSC-derived neurons. The authors should replace "in-intro i-Neurons" with a more standard terminology such as "iPSC-derived neurons".
Author Response
Summary
The authors generated iPSCs and iPSC-derived neurons from the PBMC of multiple Rett Syndrome patients containing different MECP2 mutations, and comprehensively characterized the morphological defects and electrophysiological abnormalities of RTT neurons, showing that the strength of cellular phenotypes correlates with the clinical severity of the human patients.
This work has two major contributions:
- One is the systematic generation of a collection of iPSC-derived RTT neurons from both male and female patients, with 3 different genetic mutations causing either severe or mild clinical symptoms. The authors also generated isogenic controls by leveraging balanced XCI cell lines, making the phenotype evaluation more precise. This set of cell lines can be valuable for future research on RTT and MeCP2.
- The second contribution is the evaluation of cellular phenotypes in iPSC-derived RTT neurons, including morphological measurements and electrophysiological analysis. The phenotypes of nuclear size, branch number, and immature electrical properties are largely expected from these neurons based on previous findings, but it is great that the authors establish a correlation between phenotype strength and clinical severity of the patients, providing a useful biomarker moving forward.
Overall, this work is worth publishing given the contributions of creating multiple new iPSC lines for studying 3 different genetic mutations in RTT neurons, and the phenotypic validation of these lines. Before publishing, I'd like the authors to address the comments below (some are marked optional).
General Comments
- [optional improvement] The main limitation of this work are: (1) the lack of new scientific discoveries or mechanistic insights, and (2) the authors did not show evidence that these iPSC-derived neurons can be used for effective/reliable drug testing (e.g. if we can see robust results in these neurons following drug treatments). More data on either one of these directions can significantly improve the strength of this manuscript. Given the strengths of this manuscript, improving upon these limitations is optional.
We thank the reviewer for considering the balance of strengths and limitations of our manuscript pending in favor of the strengths. We fully agree on exploiting the iPSC-derived neurons as platform for drug testing with a translational scope. The long term in vitro culture required to differentiate neurons from multiple patients, amplified by the number of clones to be followed-up in parallel is a demanding task, hard to be pursued together with the characterization of neurons’ morphological and functional biomarkers. The reviewer has perfectly designed our ongoing research programme !
- The authors often use the word "altered" to describe the changes in electrophysiological properties (e.g. cell capacitance, membrane resistance). When possible (i.e. the referred changes are in one direction), these descriptions should be made more precise by saying whether it's increased or decreased.
In the Results we changed “altered” in the precise direction of the change.
- In Figure 6, Pt A2's mutant strain was used in two analysis (Fig 6b/c and Fig 6d/e) but the results are quite different. In 6c, the Pt A2 Mut has reduced membrane potential and AP threshold compared to control, but in 6e, both phenotypes are in the increasing direction. While variability between different sets of trials is understandable, I'd expect the change to be at least in the same direction. Having them in different directions suggests there are a lot of variabilities in the experimental protocol.
We have recapitulated for the reviewer the findings of Fig. 6b/c and Fig 6d/e for PtA2:
vs pooled controls vs isogenic control
Cell capacitance reduced reduced
Membrane resistance not different increased
Membrane potential not different increased
AP threshold reduced not different
One parameter is similarly regulated, while the other three parameters the differences that were present in one comparison were not present in the other condition, but there is no opposite effect.
Specific Comments
- The word "i-Neurons" in the paper title is confusing, and not a common way to refer to iPSC-derived neurons. The authors should replace "in-intro i-Neurons" with a more standard terminology such as "iPSC-derived neurons".
We replaced iPSC neurons with “IPSC-derived neurons” as suggested